

# Association between plain ropivacaine dose and spinal hypotension for cesarean delivery: a retrospective study

Min Li[1,2], Guohao Xie[1], Lihua Chu[1] and Xiangming Fang[1]

[1] Department of Anesthesiology, Zhejiang University School of Medicine First Affiliated Hospital, Hangzhou, Zhejiang, China
[2] Department of Anesthesiology, The First Hospital of Fuyang, Hangzhou, Hangzhou, Zhejiang, China

## ABSTRACT

**Background**. Data on the association between the plain ropivacaine dose and maternal hypotension during cesarean delivery are limited. Thus, this study aimed to explore this association.

**Methods**. This retrospective study included patients undergoing cesarean sections under spinal or combined spinal-epidural anesthesia with plain ropivacaine at The First Hospital of Fuyang, Hangzhou, China, between 2018 and 2022. Data were obtained from the anesthesia information management system. Liner trend tests were used to distinguish the linear relationship between spinal hypotension and the plain ropivacaine dose, and receiver operating characteristic curves were used to calculate the dose threshold. Logistic regression was used to adjust for confounders. Sensitivity analyses were performed to evaluate the stability of the results. The secondary outcome was vasopressor use (metaraminol and ephedrine).

**Results**. In total, 1,219 women were included. The incidence of hypotension linearly correlated with the plain ropivacaine dose (adjusted $P$-value for trend, $P < 0.001$). Thus, we used a dose threshold of 17.5 mg to compare the dose as a binary variable ($\geq 17.5$ mg $vs.$ $< 17.5$ mg). Plain ropivacaine doses of $\geq 17.5$ mg were associated with a higher incidence of spinal hypotension (adjusted odds ratio: 2.71; 95% confidence interval [1.85–3.95]; $P < 0.001$). The sensitivity analyses yielded similar results. The plain ropivacaine dose also correlated with metaraminol use but not ephedrine use.

**Conclusions**. The incidence of spinal-induced hypotension in women undergoing cesarean section linearly correlated with the plain ropivacaine dose. The dose threshold for hypotension risk was 17.5 mg.

## INTRODUCTION

Cesarean section is mostly performed under spinal anesthesia. Ropivacaine is a well-tolerated anesthetic with low central nervous system toxicity and cardiotoxicity (*Simpson et al., 2005*) that has long been used in cesarean delivery through intrathecal anesthesia. In recent years, plain ropivacaine has proved unreliable and unpredictable compared to hyperbaric ropivacaine with glucose, (*Fettes et al., 2005*; *Hocking & Wildsmith, 2004*; *van Kleef, Veering & Burm, 1994*) requiring anesthesiologists to become "bartenders" during

Corresponding author
Xiangming Fang, xmfang@zju.edu.cn

cesarean sections, introducing glucose (*Khaw et al., 2002*) or adjuvants, such as sufentanil, (*Gautier et al., 2003*) fentanyl, or morphine, (*Uppal et al., 2020*) as needed. However, every extra medication injected into the spine increases the risk to the patient owing to the possibility of bacterial, particle, or other contaminations, dosage errors, or incorrect medication administration. Moreover, extra medications require additional time and expense (*Tulchinsky, 2020*), and it did not show any advantage over plain anesthetics in terms of hypotension (*Sng et al., 2018*; *Uppal et al., 2020*). Therefore, optimizing the administration of plain ropivacaine should remain a priority.

The effective dose of plain ropivacaine for spinal anesthesia during cesarean sections is 14.22 to 26.8 mg (*Khaw et al., 2001*; *Parpaglioni et al., 2006*). Hypotension after cesarean section with spinal anesthesia is a common complication (*Campbell & Stocks, 2018*; *Zieleskiewicz et al., 2018*). The accepted definition of spinal hypotension is a systolic blood pressure decrease >20% from the baseline value or systolic blood pressure of <100 mmHg (*Kinsella et al., 2018*). Few studies exist on the association between the plain ropivacaine dose and the incidence of spinal hypotension after cesarean section, and they only describe the incidence as a secondary outcome (*Khaw et al., 2001*; *Khaw et al., 2002*; *Parpaglioni et al., 2006*). Therefore, this single-center retrospective study explored this relationship.

## METHODS

### Ethical approval and clinical trial registration
The Institutional Review Board of the First Hospital of Fuyang, Hangzhou, China, authorized this study on December 19, 2022 (Ethical approval number: 2022-lw-034) and waived the written informed consent requirement owing to the retrospective nature of the study design. The trial was registered on Chinese Clinical Trial Registry (ChiCTR2300071440, Principal investigator: Min Li, Date of registration: 16th May 2023). Data collection started on May 17, 2023. This study followed the STROBE (Strengthening the reporting of observational studies in epidemiological guidelines) statement (*Von Elm et al., 2007*).

### Background
This retrospective study was conducted in a tertiary medical center (the First People's Hospital of Fuyang, Hangzhou, China). Data was collected from the Anesthesia Information Management System (AIMS) from January 2018 to December 2022.

### Inclusion criteria
Patients who underwent cesarean delivery and received single spinal anesthesia or combined spinal-epidural anesthesia (CSE) with 0.75% ropivacaine without any adjunct, blood pressure, and plain ropivacaine dose were included.

### Exclusion criteria
Patients administered general anesthesia or epidural anesthesia during the cesarean delivery, or those who received CSE and epidural supplements were excluded.

### Nerve axis anesthesia

Spinal anesthesia was administered in the lateral position. After infiltration with 2% lidocaine (2–3 mL), a Quincke needle (22-gauge for spinal anesthesia; Zhejiang Fert Medical Equipment Co., Ltd., China) was used to locate the intrathecal space. The anesthetic was 0.75% ropivacaine alone, and the anesthesiologists themselves determined the dosage. After the intrathecal injection, the patient was placed in the left lateral tilt position (bed tilted 15–30°) until the surgeons indicated the posture requirements. The operation began after reaching the T7 touch block height. When conducting CSE, the anesthesiologist used an 18-gauge Touhy needle and a 25-gauge, 110 mm pencil-point spinal needle (Zhejiang Fert Medical Equipment Co., Ltd., China). No prophylactic vasopressors were administered during the study period. Metaraminol was the first choice, and ephedrine was recommended if the maternal heart rate was less than 50 beats/min.

### Data and sources

The following information was obtained from the AIMS: maternal age, maternal weight, hypertension disorder complicating pregnancy, singleton or multiple pregnancies, gestational age, anesthesia to incision time, anesthesia to delivery time, single spinal anesthesia or CSE, anesthesia puncture site (L2/3, L3/4), anesthesiologist seniority, planned or emergency surgery, and vasopressor use (metaraminol and ephedrine).

The primary variable was the dose of plain ropivacaine administered, and the primary outcome was the frequency of spinal hypotension. If the systolic blood pressure dropped below the baseline value by >20% or <100 mmHg, then the situation was recorded (*Klöhr et al., 2010*). The secondary outcome was vasopressor use (metaraminol and ephedrine).

### Statistical analyses

Continuous variables were presented as medians and interquartile ranges and compared using the Kruskal–Wallis test after checking for normality. Classified variables were presented as counts and percentages and compared with the $\chi^2$ or Fisher's exact tests. To identify the linear trend relationship between incident hypotension and the plain ropivacaine dose and to increase the ability of the model to detect risk, the plain ropivacaine doses were divided into quartile groups: Q1 ($\leq$15.75 mg), Q2 (15.76–16.50 mg), Q3 (16.51–17.25 mg), and Q4 ($\geq$17.26 mg) (*Chen et al., 2024*; *Currenti et al., 2023*; *Zheng et al., 2023*). Each median value of the plain ropivacaine dose (Q1–Q4) was assigned to rank the quartile groups, and the *P*-value (*P* for trend) was calculated using logistic regression. To ensure a linear correlation, dummy variables were created for the quartile groups, and the Q1 group was defined as the reference for comparison. Multivariate models (1 and 2) were adjusted for confounding factors.

The threshold ropivacaine dose was evaluated using receiver operating characteristic (ROC) curves extracted from the univariate logistic regression model (*Weiniger et al., 2021*). Subsequently, we collapsed the ropivacaine doses into binary categorical variables (above *vs.* below the threshold) for further analyses, then conducted a univariate analysis to determine the potential confounding factors related to spinal hypotension. A multivariate logistic regression analysis was used to characterize the correlation between the incidence of spinal hypotension and the binary plain ropivacaine doses.

Sensitivity analyses were also performed. First, we used multiple imputations for missing data using the variables considered for the regression analysis (*Tessler et al., 2018*). Reports indicate that pregnant patients with hypertension under spinal anesthesia were less likely to develop hypotension; thus, the association was re-analyzed by excluding patients with hypertension.

The secondary outcome was vasopressor use (metaraminol and ephedrine). Spearman's rank correlation was used to evaluate the relationship between the plain ropivacaine dose and vasopressor use.

A sample size calculation was not performed; all available data was analyzed. Statistical analyses were performed using SPSS version 26 (IBM Corp., Armonk, NY, USA). *P*-values of <0.05 were considered statistically significant.

## RESULTS

### Patient characteristics

In total, 1,219 women were included. Figure 1 and Tables 1 and 2 summarize the patient selection process and their characteristics. Overall, 854 (72.3%) patients experienced hypotension, 606 (51.2%) received metaraminol, 151 (12.8%) received ephedrine, and 54 (4.6%) received both vasopressors.

### Plain ropivacaine dose and hypotension

In total, 312 (26.4%), 408 (34.5%), 181 (15.3%), and 281 (23.8%) patients were categorized into the Q1–Q4 plain ropivacaine dose quartiles, respectively (Table 3). Q4 significantly differed from Q1 (odds ratio (OR): 2.72 95% confidence interval (CI) [1.81–4.09]; $P < 0.001$). Thus, the median values of each quartile were used for the trend test, which demonstrated a linear relationship between the spinal hypotension incidence and the plain ropivacaine dose (*P*-value for trend < 0.001; adjusted *P*-value < 0.001; Table 3 and Fig. S1).

Next, the plain ropivacaine dose was classified as a binary variable based on the threshold established from the ROC curve in the univariate logistic regression model; the plain ropivacaine threshold was 17.5 mg. The incidence of spinal hypotension was higher in the ≥17.5 mg group than in the <17.5 mg group (85.4% *vs.* 68.1%, $P < 0.001$). After adjusting for confounding factors, the adjusted OR was 2.71 (95% CI [1.85–3.95]; $P < 0.001$; Table 4). The area under the ROC curve was 0.5748, suggesting the model could distinguish between patients with and without hypotension (Fig. S2).

### Sensitivity analyses

The linear relationship between the incidence of spinal hypotension and the plain ropivacaine dose appeared stable. Multiple imputations for the missing data and other groups yielded similar results (Table S1). We also performed multiple imputations to compare the binary variable (≥17.5 mg *vs.* <17.5 mg of ropivacaine), which produced similar results (OR: 2.64; 95% CI [1.82–3.82], $P < 0.001$).

### Vasopressor correlations

The plain ropivacaine dose correlated with metaraminol use but not ephedrine use (Spearman's rank correlation coefficient 0.07, $P < 0.001$ *vs.* 0.02, $P = 0.58$).

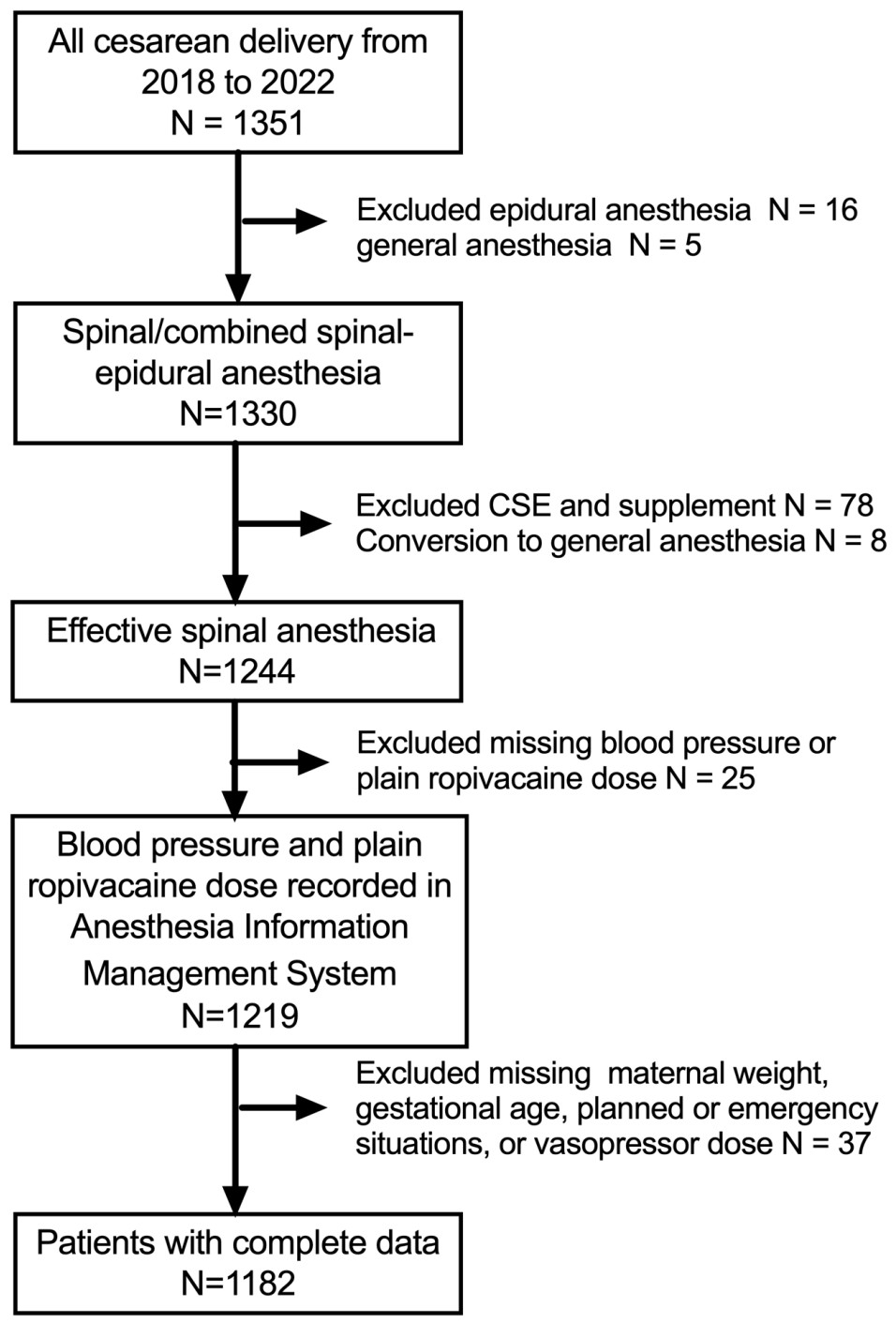

**Figure 1 Patient selection flow chart.**

## DISCUSSION

This retrospective study identified a linear association between the incidence of spinal hypotension and the plain ropivacaine dose, with a cutoff value of 17.5 mg. Spinal-induced

**Table 1  Characteristics of anesthesia, maternal details according to dose of plain ropivacaine at baseline (N = 1,219).**

| Characteristic | All cohort (N = 1,219)[e] | Q1[a] n = 312 | Q2[a] n = 408 | Q3[a] n = 181 | Q4[a] n = 281 | P value |
|---|---|---|---|---|---|---|
| Maternal age, y, median [IQR][b] | 34.0 [30.0,37.0] | 34.0 [30.0,37.0] | 33.0 [30.0,36.0] | 34.0 [31.0,38.0] | 34.0 [30.0,37.0] | 0.06 |
| Maternal weight, kg, median [IQR][b] | 70.0 [64.8,76.0] | 70.0 [64.2,76.0] | 70.0 [65.0,76.6] | 70.0 [64.0,76.1] | 70.0 [65.0,76.2] | 0.87 |
| Gestation age, week, median [IQR][b] | 39.0 [38.0,39.0] | 38.0 [38.0,39.0] | 39.0 [38.0,39.0] | 38.7 [38.0,39.0] | 39.0 [38.0,39.0] | <0.001 |
| Anesthesia mode, n (%)[d] | | | | | | 0.195 |
| Spinal | 17 (1.4) | 3 (0.9) | 3 (0.7) | 4 (2.1) | 7 (2.4) | |
| Combined spinal-epidural (CSE) | 1202 (98.6) | 315 (99.1) | 414 (99.3) | 191 (97.9) | 282 (97.6) | |
| Puncture site, n (%)[c] | | | | | | <0.001 |
| L2/3 | 746 (61.2) | 220 (69.2) | 180 (43.2) | 59 (30.3) | 136 (47.1) | |
| L3/4 | 473 (38.8) | 98 (30.8) | 237 (56.8) | 136 (69.7) | 153 (52.9) | |
| Planned cesarean delivery, n (%)[c] | 583 (48.8) | 142 (45.2) | 198 (47.8) | 98 (53.8) | 145 (50.9) | 0.25 |
| Seniority of anesthesiologists, n (%)[c] | | | | | | <0.001 |
| Resident | 362 (29.7) | 84 (26.4) | 71 (17.0) | 55 (28.2) | 152 (52.6) | |
| Attending | 373 (30.6) | 118 (37.1) | 178 (42.7) | 21 (10.8) | 56 (19.4) | |
| Resident and attending | 484 (39.7) | 116 (36.5) | 168 (40.3) | 119 (61.0) | 81 (28.0) | |
| Hypertensive disease, n (%)[c] | 57 (4.7) | 16 (5.0) | 16 (3.8) | 6 (3.1) | 19 (6.6) | 0.24 |
| Multiple gestation, n (%)[c] | 27 (2.2) | 14 (4.4) | 7 (1.7) | 1 (0.5) | 5 (1.7) | 0.02 |
| Single gestation, n (%)[c] | 1192 (97.8) | 304 (95.6) | 410 (98.3) | 194 (99.5) | 284 (98.3) | |

**Notes.**
[a]Q1, Q2, Q3, Q4 are quartiles of plain ropivacaine dose (mg, median (range)).
[b]Nonparametric test (Kruskal–Wallis test).
[c]$\chi^2$ test.
[d]Fisher exact test.
[e]N noted where different.
Abbreviation: IQR, interquartile range; SD, standard deviation.

**Table 2  Anesthesia details according to dose of plain ropivacaine at baseline (N = 1,219).**

| Characteristic | All cohort (N = 1,219)[d] | Q1[a] n = 312 | Q2[a] n = 408 | Q3[a] n = 181 | Q4[a] n = 281 | P value |
|---|---|---|---|---|---|---|
| Ropivacaine dose, mg, median [IQR][b] | 16.5 (15.75,17.25) | 15.0 (13.5,15.75) | 16.5 (16.5,16.5) | 17.25 (17.25,17.25) | 18.75 (18.0,18.75) | |
| Hypotension occurred, n (%)[c] | 877 (71.9) | 216 (67.8) | 289 (69.3) | 126 (64.6) | 246 (85.1) | <0.001 |
| Metaraminol, dose, mg, median [IQR][b] | 0 (0, 0.5) | 0 (0.5) | 0.25 (0, 0.5) | 0 (0, 0.5) | 0.5 (0, 0.5) | <0.001 |
| Ephedrine, dose, mg, median [IQR][b] | 0 (0, 6) | 0 (0, 6) | 0 (0, 0) | 0 (0, 6) | 0 (0, 0) | 0.16 |
| Anesthesia to incision time, min, median [IQR][b] | 11 (9, 15) | 12 (10, 16) | 10 (8, 15) | 11 (9,15) | 11 (8, 15) | 0.06 |
| Anesthesia to delivery time, min, median [IQR][b] | 19 (16, 24) | 19 (15, 23) | 18 (15, 23) | 18 (15, 23) | 18 (15, 23) | 0.04 |

**Notes.**
[a]Q1, Q2, Q3, Q4 are quartiles of plain ropivacaine dose (mg, median (range)).
[b]Nonparametric test (Kruskal–Wallis test).
[c]$\chi^2$ test.
[d]N noted where different.
Abbreviation: IQR, interquartile range; SD, standard deviation.

**Table 3** Association between plain ropivacaine dose and incident hypotension ($N = 1,182$).

| | OR (95 CI) | | | | P value for trend |
| | Q1[a] $n = 312$ | Q2[a] $n = 408$ | Q3[a] $n = 181$ | Q4[a] $n = 281$ | |
|---|---|---|---|---|---|
| Crude | 1.0 | 1.05 (0.77, 1.45) | 0.87 (0.59, 1.28) | 2.72 (1.81, 4.09) | <0.001 |
| P values[*] | | 0.75 | 0.48 | <0.001 | |
| Model 1[b] | 1.0 | 1.18 (0.85, 1.64) | 0.91 (0.607, 1.36) | 2.85 (1.85, 4.37) | <0.001 |
| P values[*] | | 0.32 | 0.64 | <0.001 | |
| Model 2[c] | 1.0 | 1.19 (0.85, 1.65) | 0.89 (0.59, 1.34) | 2.83 (1.84, 4.36) | <0.001 |
| P values[*] | | 0.31 | 0.58 | <0.001 | |

Notes.

[a]Q1, Q2, Q3, Q4 are quartiles of plain ropivacaine dose (mg, median (range)).

[b]Model 1 was adjusted for maternal age, hypertension, planned or emergency surgery, puncture location (L2/3, L3/4), anesthesiologist seniority, anesthesia to delivery time.

[c]Model 2 was adjusted for maternal age, maternal weight, gestational age, hypertension, singleton or multiple pregnancies, planned or emergency surgery, puncture location (L2/3, L3/4), anesthesiologist seniority, anesthesia to incision time, and anesthesia to delivery time.

[*]Q2, Q3, and Q4 are all compared to Q1.

Abbreviation: OR, Odds ratio; CI, Confidence interval.

**Table 4** Multivariable logistic regression model of plain ropivacaine dose effect on spinal-induced hypotension adjusted for confounders ($N = 1,182$).

| Characteristic | Univariate analysis | | Multivariate analysis | |
|---|---|---|---|---|
| | OR (95% CI) | P value | OR (95% CI) | P value |
| Plain ropivacaine dose $\geq$17.5 mg vs <17.5 mg | 2.74 (1.91–3.92) | <0.001 | 2.71 (1.85 to 3.95) | <0.001 |
| Maternal age, y, (continuous) | 1.02 (0.99–1.05) | 0.17 | | |
| Maternal weight, kg, (continuous) | 1.01 (0.995–1.02) | 0.25 | | |
| Gestation age, week, (continuous) | 0.96 (0.91–1.02) | 0.20 | 0.95 (0.89–1.01) | 0.12 |
| Hypertensive disease | 0.51 (0.25–1.05) | 0.07 | 0.53 (0.25–1.11) | 0.09 |
| Single vs Multiple gestation | 1.10 (0.48–2.54) | 0.83 | | |
| Planned vs Emergency | 1.03 (1.01–1.68) | 0.04 | 0.82 (0.63–1.06) | 0.13 |
| CSE vs spinal | 1.57 (0.57–4.36) | 0.38 | | |
| Anesthesia to incision time, min, (continuous) | 1.01 (0.99–1.04) | 0.32 | | |
| Anesthesia to delivery time, min, (continuous) | 1.01 (0.99–1.03) | 0.30 | 1.01 (0.99–1.03) | 0.45 |
| L3/4 vs L2/3 | 0.64 (0.50–0.83) | 0.001 | 0.56 (0.43–0.74) | <0.001 |
| Attending vs resident | 0.52 (0.38–0.72) | <0.001 | 0.64 (0.45–0.90) | 0.009 |
| Attending and resident vs attending | 1.12 (0.83–1.51) | 0.45 | | |
| Attending and resident vs resident | 0.59 (0.42–0.83) | 0.002 | 0.80 (0.56–1.15) | 0.23 |

Notes.

Abbreviation: OR, Odds ratio; CI, Confidence interval; vs, versus.

hypotension is mainly caused by a decrease in efferent sympathetic outflow (*Kinsella et al., 2018*), which is related to the local anesthesia dose. The incidence of spinal-induced hypotension can be reduced by using smaller amounts of anesthetics. Our findings are consistent with previous studies that focused on bupivacaine anesthetics (*Arzola & Wieczorek, 2011*; *Ben-David et al., 2000*; *Leo et al., 2009*; *Van de Velde et al., 2006*; *Weiniger et al., 2021*) and address the knowledge gap regarding plain ropivacaine dose and incident hypotension.

## Sensitivity analysis

Multiple imputations were performed for missing values in the sensitivity analysis. Women with preeclampsia have a lower incidence of hypotension after spinal anesthesia than those without (*Aya et al., 2003*; *Kinsella et al., 2018*). Thus, we excluded patients with pregnancy-induced hypertension from the re-analysis to minimize the effects of gestational hypertension. The OR values did not differ significantly between the original and re-analyses, confirming our results.

## Comparisons with previous studies

Few studies have focused on the association between the plain ropivacaine dose and spinal hypotension during cesarean delivery. Glucose-free bupivacaine, like plain ropivacaine, is an extensively studied hypobaric anesthetic with conflicting results regarding its dose effects on hemodynamics (*Arzola & Wieczorek, 2011*; *Ben-David et al., 2000*; *Bryson et al., 2007*; *Carvalho et al., 2005*). In contrast, there is a consensus that the amount of hyperbaric anesthetic affects hypotension (*Kiran & Singal, 2002*; *Leo et al., 2009*; *Tessler et al., 2018*; *Van de Velde et al., 2006*), since a small dose of hyperbaric drug used intrathecally during a cesarean section offers a reliable cephalad spread, reflected in a lower incidence of hypotension (*Vercauteren et al., 1998*) due to the influence of gravity and the curves of the vertebral column. However, hypobaric anesthetics, such as plain ropivacaine, are unpredictable due to the absence of gravity, and the effects vary with the addition of glucose, which changes the viscosity. Moreover, they are limited by the influence of the spine's anatomy and the operating technique owing to baricity, which changes with temperature (*McLeod, 2004*); thus, its movement changes (*i.e.,* it sinks or floats) based on the body's position. Nonetheless, we identified a linear relationship between plain ropivacaine use and incident hypotension. Our secondary outcome was also consistent with previous studies that reported an association between the plain ropivacaine dose and metaraminol use but not ephedrine use (*Ikeda et al., 2023*).

## Mechanisms

Spinal-induced hypotension, with a deep and extensive sympathetic block on an etiological basis, is a direct consequence of local anesthetic administration to the nerve axis. Therefore, the local anesthetic dose is manipulated to reduce the incidence and severity of hypotension, aiming for the smallest possible "appropriate dose" to produce the "optimal" anesthetic effect (*Benhamou & Wong, 2009*).

Clinically, however, changes in maternal hemodynamics during spinal anesthesia with plain ropivacaine are more susceptible to manipulation techniques such as injection

temperature, lumbar segment, injection position (sitting/lateral), and injection speed, among other interventions, such as fluid load, bed tilt, and opioid adjuvants. These manipulations may weaken or mask the effects of the drug dose (*Chooi et al., 2020*; *Stienstra & Van Poorten, 1988*; *Stienstra & Veering, 1998*). For plain ropivacaine, these effects are related to the baricity of ropivacaine. Plain ropivacaine is hypobaric at body temperature, even in full-term pregnant women (the density of 0.75% plain ropivacaine is 0.9953 g/mL, (*McLeod, 2004*) and the density of cerebrospinal fluid (CSF) in full-term pregnant women is 1.00030 (0.00004) g/mL at body temperature (*Richardson & Wissler, 1996*)). However, the density of the local anesthesia solution decreases as temperature increases, making its diffusion motion more complex. Plain ropivacaine is hyperbaric outside the body, especially at room temperature or below, but its density and viscosity decrease within a few minutes of entering the subarachnoid space. Therefore, plain ropivacaine will sink in the CSF based on the patient's posture. Adaptation to the CSF temperature will slow its deposition rate until it becomes isobaric (34.8 °C) and then hypotensive, subsequently rising and floating again (*Heller et al., 2006*). Despite these factors, our study showed a linear relationship between the dose and hypotension and identified doses associated with a higher risk of hypotension.

Hypotension with bradycardia is a rare vascular vagal response pattern that is not directly proportional to the height of the blockade (*Kinsella & Tuckey, 2001*), which might explain why we found an association between the plain ropivacaine dose and metaraminol but not ephedrine. A series of sudden bradycardia caused by activation of the vasovagal nerve (also known as Bezold-Jarisch) reflex is well established in obstetric anesthesia. This change is triggered by the mechanism of reduced cardiac venous return, which may occur during regional block, bleeding, or supine inferior vena cava compression during pregnancy, and these factors combined are additive. The possible reason is that inferior vena cava compression is not taken seriously. This risk exists even in the absence of sympathetic block from regional anesthesia.

However, the degree of hypotension is directly related to the number of blocked preganglionic sympathetic segments, and even if not blocked, reducing preload (by compressing the inferior vena cava) (*Shayegan, Khorasani & Knezevic, 2018*) or reducing systemic vascular resistance (oxytocin and cabetoxin) (*Langesaeter, Rosseland & Stubhaug, 2009*; *Rosseland et al., 2013*) is a significant risk factor for developing hypotension. Therefore, low-dose ropivacaine in this study can reduce the occurrence of hypotension and the demand for vasopressor, but the incidence of hypotension is still around 70%, and the correlation with vasopressors is relatively weak.

## Limitations

This study has several limitations. First, we lacked comprehensive data about concurrent diseases, such as hypertensive disorders complicating pregnancy, the reason for emergency surgery, anesthesia details (*e.g.*, the block level), and other factors (*e.g.*, the injection temperature, injection speed, CSF volume pumped back, and placement time after injecting the spinal anesthesia drugs). Second, this retrospective study was a single-center trial with a limited sample size; thus, it may not be generalizable. Third, the first available blood

pressure measurement in the AIMS system was considered the baseline value instead of using three separate measurements. Nonetheless, our study included a large number of patients and had good control of the confounding factors, which strengthened our results. Additionally, we conducted a series of sensitivity analyses to examine the stability of the outcomes. Finally, the vasopressors were not administered prophylactically, providing an opportunity to investigate these relationships.

## CONCLUSIONS

This study identified a linear relationship between the incidence of spinal-induced hypotension and the plain ropivacaine dose during cesarean section. Doses above ≥17.5 mg were associated with a higher incidence of spinal hypotension. In addition, the dose of plain ropivacaine correlates with the dose of vasopressor required. However, both groups had a sufficiently high incidence of spinal hypotension that titration vasopressor therapy was unavoidable.

## ACKNOWLEDGEMENTS

The authors thanks to the staff of the Department of Anesthesiology, the First Hospital of Fuyang, Hangzhou, China for data extraction.

### Funding
The authors received no funding for this work.

### Competing Interests
The authors declare there are no competing interests.

### Author Contributions
- Min Li conceived and designed the experiments, performed the experiments, prepared figures and/or tables, authored or reviewed drafts of the article, and approved the final draft.
- Guohao Xie analyzed the data, prepared figures and/or tables, authored or reviewed drafts of the article, and approved the final draft.
- Lihua Chu analyzed the data, authored or reviewed drafts of the article, and approved the final draft.
- Xiangming Fang analyzed the data, authored or reviewed drafts of the article, and approved the final draft.

### Human Ethics
The following information was supplied relating to ethical approvals (i.e., approving body and any reference numbers):

The Institutional Review Board of the First Hospital of Fuyang, Hangzhou, China, authorized this study on December 19, 2022 (Ethical approval number: 2022-lw-034).

## Clinical Trial Ethics

The following information was supplied relating to ethical approvals (i.e., approving body and any reference numbers):

The Institutional Review Board of the First Hospital of Fuyang, Hangzhou, China, authorized this study.

## Data Availability

The raw measurements are available in the Supplementary File.

## Clinical Trial Registration

The following information was supplied regarding Clinical Trial registration:

ChiCTR2300071440

## Supplemental Information

Supplemental information for this article can be found online at http://dx.doi.org/10.7717/peerj.18398#supplemental-information.

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
