# Peer review of "Association between plain ropivacaine dose and spinal hypotension for cesarean delivery: a retrospective study"

_PeerJ, doi:10.7717/peerj.18398_

## Round 0.1 · original submission · Major Revisions

In particular, the comments of R2 must be addressed adequately.

Reviewer 1 ·

Basic reporting

no comment

Experimental design

We appreciate authors for their hard work. However, it is very pity that this manuscript have several critical flaws.
1. In this study, the plain ropivacaine doses were divided into quartile groups: Q1 (≤15.75 mg), Q2 (15.76-16.50 mg), Q3 (16.51-17.25 mg), and Q4 (≥17.26 mg). As far as I can gather, it is hard to use the dose because the dose of 15.75mg ropivacaine was 2.1ml and the dose of 16.5mg was 2.2 ml. Please detail the procedure and basis. How to determine the usage between 15.76-16.50 mg.
2. The height should be identified as the key influence during spinal or combined spinal-epidural anesthesia.
3. Maternal left-lateral position was usually used during cesarean section. How about some patients use of left-lateral position? How about the rates of left-lateral position?

Validity of the findings

no comment

Reviewer 2 ·

Basic reporting

no comment

Experimental design

“Women with preeclampsia have a lower incidence of hypotension after spinal anesthesia than those without.(Aya et al. 2003; Kinsella et al. 2018) Thus, we excluded patients with pregnancy-induced hypertension from the re-analysis to minimize the effects of gestational hypertension.” Why the exclusion criteria did not include gestational hypertension?

Validity of the findings

no comment

Additional comments

Thank you very much for inviting me to review this article. I noticed that the study has several issues. Given this unreliable statistic, I do not believe that the conclusion is also reliable. The current manuscript cannot be accepted for publication.
Methods
1. “Women with preeclampsia have a lower incidence of hypotension after spinal anesthesia than those without.(Aya et al. 2003; Kinsella et al. 2018) Thus, we excluded patients with pregnancy-induced hypertension from the re-analysis to minimize the effects of gestational hypertension.” Why the exclusion criteria did not include gestational hypertension?

2.“The operation began after reaching the T7 sensation block height.” Whether the sensory block plane is inadequate? How to control the same level of sensation of T7 in a retrospective study? Why chose the block level of T7 as an included criterion? Why not a T6, T5 or T4 block? According to previous studies, a cold sensory level of T5 or higher is required to satisfy cesarean delivery[ Ousley R, Egan C, Dowling K, Cyna AM. Assessment of block height for satisfactory spinal anaesthesia for caesarean section. Anaesthesia. 2012. 67(12): 1356-63.].

3.Provide references to the primary outcome was the frequency of spinal hypotension. If the systolic blood pressure dropped below the baseline value by >20% or <100 mmHg.

4.Why the plain ropivacaine doses were divided into quartile groups: Q1, Q2, Q, and Q4?

5.Statistics is the biggest problem, there are many factors that affect blood pressure during cesarean section surgery. Have they been comprehensively considered and included in the analysis? There are many confounding factors in retrospective analysis, and how to control these confounding factors is the main basis for determining the reliability of the conclusion.
Anesthesia to incision time, Anesthesia to delivery time, Planned vs Emergency, Attending vs resident, you have included the above factors in the regression equation analysis, are these important factors that affect blood pressure? Is there any reference basis?

Discussion
“Hypotension with bradycardia is a rare vascular vagal response pattern that is not directly proportional to the height of the blockade,(Kinsella & Tuckey 2001) which might explain why we found an association between the plain ropivacaine dose and metaraminol but not ephedrine. ” Please clarify

Conclusion
This article describes the dose correlation between generic ropivacaine doses and vasopressors. Why the plain ropivacaine dose influences the required prophylactic dose of the vasopressor in conclusions.

---

## Round 0.2 · accepted · Accept

I confirmed that the authors have addressed all the reviewers' comments.

Reviewer 1 ·

Basic reporting

no coment

Experimental design

no comment

Validity of the findings

no comment

Reviewer 2 ·

Basic reporting

no comment

Experimental design

no comment

Validity of the findings

no comment

Additional comments

no comment